

# *Porphyromonas gingivalis* lipopolysaccharide promotes T-hel per17 cell differentiation by upregulating Delta-like ligand 4 expression on CD14⁺ monocytes

Chi Zhang[1,2], Chenrong Xu[3], Li Gao[1,2], Xiting Li[1,2] and Chuanjiang Zhao[1,2]

[1] Department of Periodontology, Hospital of Stomatology, Sun Yat-sen University, Guangzhou, China
[2] Guangdong Provincial Key Laboratory of Stomatology, Guanghua School of Stomatology, Sun Yat-sen University, Guangzhou, China
[3] Department of Periodontology, Guangdong Provincial Hospital of Stomatology, Stomatological Hospital of Southern Medical University, Guangzhou, China

## ABSTRACT

**Backgroud**. To investigate the effect and mechanism of *Porphyromonas gingivalis* (*P. gingivalis*) lipopolysaccharide (LPS) on Th17 cell differentiation mediated by CD14⁺ monocytes.

**Methods**. *P. gingivalis* LPS-activated CD14⁺ monocytes were co-cultured with CD4⁺T cells in different cell ratios. An indirect co-culture system was also established using transwell chambers. Furthermore, anti- Delta-like ligand 4 (Dll-4) antibody was used to investigate the role of Dll-4 in Th17 cell response. The mRNA expression was analyzed using quantitative reverse transcription-polymerase chain reaction, and secreted cytokines in culture supernatant were detected using enzyme-linked immunosorbent assay. Flow cytometry was used to determine the frequencies of Th17 cells. IL-17 protein expression levels were determined using western blotting assay.

**Results**. *P. gingivalis* LPS increased the expressions of interleukin (IL)-1$\beta$, IL-6, IL-23 and transforming growth factor (TGF)-$\beta$ in CD14⁺ monocytes. Th17 cell frequency upregulated, which is not solely cytokine-dependent but rather requires cell-cell contact with activated monocytes, particularly in the 1:10 cell ratio. Furthermore, *P. gingivalis* LPS increased t he expression of Dll-4 on CD14⁺ monocytes, whereas the anti- Dll-4 a ntibody decreased the response of Th17 cells. The results suggest that *P. gingivalis* LPS enhances Th17 cell response via Dll-4 upregulation on CD14⁺ monocytes.

## INTRODUCTION

Periodontitis is a chronic inflammatory disease, characterized by attachment loss and alveolar bone resorption (*Di Benedetto et al., 2013*). It is caused by the accumulation of pathogenic microorganisms on the teeth, which stimulate local inflammatory and immune reactions. Although periodontitis is caused by pathogenic bacteria, its progression and prognosis are highly influenced by CD4⁺T cell-mediated host immune response (*Baker et al., 2001*; *Campbell et al., 2016*). Increasing evidence has shown that T-helper 17

Corresponding author
Chuanjiang Zhao,
zhaochj@mail.sysu.edu.cn

(Th17) cells that produce interleukin (IL)-17 (also known as IL-17A) (*Park et al., 2005*; *Harrington et al., 2005*) are involved in the pathogenesis of periodontitis and alveolar bone destruction (*Adibrad et al., 2012*; *Cardoso et al., 2009*). It has been reported that the percentage of IL-17 producing CD4$^+$T cells was higher in periodontitis lesions than in healthy tissues and gingivitis lesions (*Dutzan et al., 2016*; *Okui et al., 2012*). By using biopsies from periodontitis patients, *Allam et al. (2011)* reported that the number of IL-17 positive T cells was related to the severity of periodontal inflammation. Therefore, study on the mechanism of Th17 cell differentiation in the periodontal inflammatory context is important for understanding the immunopathology of periodontitis.

In the periodontal inflammatory environment, antigen-presenting cells (APCs), such as monocytes, dendritic cells (DCs), and other lymphocytes are rapidly recruited to inflammatory infection sites to participate in immune regulation. It is well known that the adaptive immune response mediated by microorganism-activated APCs is responsible for the progression and prognosis of periodontitis (*Cheng et al., 2016*). APCs provide activated signal molecules and a series of specific cytokines that promote Th17 cell differentiation (*Novak et al., 2010*; *Gutcher & Becher, 2007*). Monocytes are often considered as DCs precursors of DCs, and exert a distinct role in the shaping of immune response (*Geissmann et al., 2008*). Several studies have confirmed that monocytes, one of the APCs, secrete IL-23, IL-6, and IL-1β, and contribute to Th17 cell differentiation (*Segura et al., 2013*; *Evans et al., 2009*).

Cell contact between monocytes and T cells plays an important role in inducing CD4$^+$T cell differentiation (*Wittmann, Alter & Stünkel, 2004*; *Roberts, Dickinson & Taams, 2015*). A recent study has shown that activated CD14$^+$ monocytes effectively induced Th17 cell differentiation only after directly contacted with CD4$^+$T cells in vitro (*Yang et al., 2017*). Another study reported that the proportion of Th17 cells in the monocytes and CD4$^+$T cell co-culture systems decreased after blocking the TNF-$\alpha$ expressed on the surface of monocytes (*Rossol, Meusch & Pierer, 2007*). Thus, the expression of intercellular adhesion molecules and co-stimulation molecules on monocytes is important in regulating the Th17 pathway.

Notch ligands and receptors expressed on CD4$^+$T cells and APCs have been found to be involved in Th17 cell differentiation (*Ito et al., 2012*; *Keerthivasan et al., 2011*). Mammals carry four Notch receptors (i.e., Notch-1, 2, 3, and 4), and five Notch ligands (i.e., Jagged-1, Jagged-2, and Delta-like 1, 3, and 4 (Dll-1, 3, 4)) (*Bray, 2006*). It has been demonstrated that Dll-4 activation of Notch may represent an important signal that instructs the development of effector T cells (*Meng et al., 2016*). Dll-4 drives Th17 cell differentiation by upregulating the transcription factor retinoid-related orphan receptor $\gamma$t (ROR $\gamma$t) in vitro (*Mukherjee et al., 2009*). In addition, DCs activation by *Escherichia coli* LPS could induce Th17 cell differentiation through upregulating the expression of Dll-4 (*Jiang et al., 2015*), suggesting a possible role of Dll-4 in modulating Th17 response elicited by periodontal pathogens.

*Porphyromonas gingivalis* is an important periodontopathic bacterium, responsible for bone resorption in periodontitis (*Hajishengallis, 2009*). *P. gingivalis*-derived lipopolysaccharide (LPS) is one of the main pathogenic factors in periodontitis and contributes significantly to the overall immune response against bacterial infection (*Jain &*

*Darveau, 2010*; *Seo et al., 2012*; *Su et al., 2015*). Our previous study showed that LPS from *P. gingivalis* directly promotes Th17 cell differentiation via toll-like receptor-2 in vitro (*Zhang et al., 2019*). In addition to the direct effect of periodontal pathogens and related virulence factors, immune cells present in the periodontal inflammatory microenvironment have also been shown to play an important role in regulating Th17 cell differentiation (*Moutsopoulos et al., 2012*). Moreover, it has been demonstrated that *P. gingivalis* activated CD14$^+$ monocytes increased IL-17 production by human CD4$^+$T cells in vitro (*Cheng et al., 2016*). However, the possible mechanism underlying the regulation of Th17 cell differentiation induced by *P. gingivalis* activated CD14$^+$ monocytes is still undetermined. Therefore, in order to gain a more comprehensive understanding of Th17 cell immunity in periodontal pathogenesis, we established cell co-culture systems of CD4$^+$T cells and *P. gingivalis* LPS-activated CD14$^+$ monocytes in vitro, to detect the Th17 cell differentiation, and further analyze the function of Dll-4 in the process.

## MATERIALS AND METHODS

### CD14$^+$ monocytes and CD4$^+$T cell sorting

Human blood samples were obtained from six healthy donors aged 18 to 23 years. The donors were recruited from the North Campus of Sun Yat-Sen University. All subjects enrolled in this study gave their informed consent before participating. This study was approved by the Ethics Committee of the Hospital of Stomatology of Sun Yat-Sen University, China (ERC-2014-08) and was conducted in accordance with the Helsiniki Declaration of 1975. Peripheral blood mononuclear cells (PBMCs) were isolated by density gradient centrifugation using Ficoll-Hypaque solution (STEMCELL, Vancouver, BC, CA) from fresh peripheral blood. CD14$^+$ monocytes (>94% purity as confirmed by flow cytometry) and CD4$^+$T cells (>96% purity) were sorted using Anti-Human CD14 Magnetic Particles-DM (BD Biosciences, San Jose, CA, USA) and a negative immunomagnetic selection for human CD4$^+$T Cell Enrichment Set-DM (BD Biosciences).

### Cell culture and stimuli

Purified CD14$^+$ monocytes and CD4$^+$T cells were cultured in RPMI 1640 medium (Life Technologies, Grand Island, NY, USA) supplemented with 10% fetal bovine serum (FBS) and 1% penicillin/streptomycin (Sigma-Aldrich, St. Louis, MO, USA), in a humidified atmosphere of 5% $CO_2$ at 37 °C. CD14$^+$ monocytes ($1 \times 10^6$/mL) were either treated for 24 h with 1 μg/mL of LPS-PG Ultrapure (InvivoGen, San Diego, CA, USA) or left untreated as controls. CD4$^+$T cells were stimulated for 24 h with plate-bound 2 μg/mL of anti-CD3 mAb and 1 μg/mL of anti-CD28 mAb (BD Biosciences).

### Co-culture and transwell assays

Activated CD4$^+$T cells ($1 \times 10^6$/mL) were cultured in 24-well plates. The CD14$^+$ monocytes treated with *P. gingivalis* LPS, or the untreated CD14$^+$ monocytes were co-cultured with the activated CD4$^+$T cells in 1:1, 1:5, or 1:10 cell ratios and were incubated for 5 days in a humidified atmosphere of 5% $CO_2$ at 37 °C. The CD14$^+$ monocytes activated by *P. gingivalis* LPS were washed 3 times with PBS to remove excess *P. gingivalis* LPS before co-cultured with CD4$^+$T cells.

### Transswell assay

For transwell assays, activated CD4$^+$T cells and CD14$^+$ monocytes were separated by an insert containing a 0.4-$\mu$m semi-permeable membrane. CD14$^+$ monocytes ($1\times 10^6$/mL) were inoculated into the upper chamber, and activated CD4$^+$T cells in a 1:10 cell ratio were inoculated into the lower chamber. The cells were cultured for 5 days in a humidified atmosphere of 5% $CO_2$ at 37 °C.

### Dll-4 blocking assay

CD14$^+$ monocytes were pretreated with 5 or 10 $\mu$g/mL anti-Dll-4 antibody (BD Biosciences) for 90 min prior to the addition of *P. gingivalis* LPS, and Mouse IgG1, $\kappa$ Isotype Control (BD Biosciences) was used as the isotype control. The cells were then incubated with 1 $\mu$g/mL of *P. gingivalis* LPS for 24 h and were co-cultured with activated CD4$^+$T cells in a 1:10 ratio for 5 days in a humidified atmosphere of 5% $CO_2$ at 37 °C. Unblocked groups were used as controls.

### RNA isolation and quantitative reverse-transcription real-time polymerase chain (qRT-PCR) reaction

RNA was extracted using TRIzol reagent (Invitrogen, Carlsbad, CA, USA), and the RNA concentration was determined using a Nanodrop 2000 spectrophotometer (Thermo Fisher Scientific, Waltham, MA, USA). Reverse transcription was performed using PrimeScript RT Master Mix (TaKaRa Biotech, Tokyo, Japan) to provide a cDNA template, and the SYBR Green I Master Kit was used to perform PCR in a Light Cycler 480 (Roche, Indianapolis, IN, USA). Table 1 lists all the primer sequences used for the target gene. $\beta$-actin was used as a loading control. The relative gene expression level was calculated using the $2^{-\Delta\Delta Ct}$ method.

### Enzyme-linked immunosorbent assay (ELISA) analysis

Secreted cytokines in culture supernatant were detected using ELISA. The concentrations of IL-1$\beta$, IL-6, IL-23 and TGF-$\beta$ in the cell culture of *P. gingivalis* LPS-treated CD14$^+$ monocytes were measured using the Quantikine Human IL-1$\beta$, IL-6, IL-23 and TGF-$\beta$ ELISA Kit (R&D system, Minneapolis, MN, USA) according to the manufacturer's instructions. The concentration of IL-17A in the co-culture of CD4$^+$T cells and CD14$^+$ monocytes was measured using the Quantikine Human IL-17A ELISA Kit (R&D system).

### Flow cytometry

For surface staining of the stimulated cells, anti-CD4-APC (BD Biosciences), anti-CD3-PERCP, anti-CD8-FITC, anti-CD14-FITC (eBioscience, San Diego, CA, USA) were used. Fc receptor binding inhibitor antibody(eBioscience) was used to inhibit the non-specific Fc-gamma receptor (FcgammaR)-mediated binding of mouse monoclonal antibodies. Fix Viability Dye (eBioscience) was used to stain and exclude dead cells from the analyses. For intracellular staining, cells were restimulated on 50 ng/mL phorbol 12-myristate 13-acetate (Sigma-Aldrich) and 250 ng/mL ionomycin (Sigma-Aldrich) in the presence of 10 $\mu$g/mL of brefeldin A (Sigma-Aldrich) for the last 5 h of culture. Cells were fixed and permeabilized using the Cytofix/Cytoperm Fixation and Permeabilization Kit (BD Biosciences), labeled

**Table 1** Primer sequences for quantitateive reverse transcription-polymerase chain reaction.

| Gene | Forward sequence(5′–3′) | Reverse sequence(5′–3′) |
|---|---|---|
| RORC | ACAGAGATAGAGCACCTGGT | CACATCTCCCACATGGACTT |
| IL-17 | ACGAAATCCGGATGCCCAA | TGCGGTGGAGATTCCAAGGT |
| Jagged-1 | ATGATGGGAACCCGATCAAG | TCACCAAGCAACAGATCCAA |
| Dll-4 | GCAAACAGCAAAACCACACA | CACACAGACTGGTACATGGA |
| IL-1β | TGAGCACCTTCTTTCCCTT | ATGGACCAGACATCACCAA |
| IL-6 | CCTTCCAAAGATGGCTGAAA | CTGGCTTGTTCCTCACTACT |
| TGF-β | CTGTAATGCTGCTGTTGCT | CTTAGATCCATGTGTGTCCCAC |
| IL-23 | GTGGAAACCCACAACGAAAT | TTTAACTTAGCCTCAGCAGA |
| β-action | TGGGACGACATGGAGAAAA | GGGGTGTTGAAGGTCTCAAA |

with anti-IL-17-PE (eBiosciences), and analyzed on the Gallios system using the Kaluza software (Beckman Coulter, CA, USA).

## Western blot analysis

Cells were collected and lysed in radioimmunoprecipitation assay (RIPA) buffer containing 1% phenylmethylsulfonyl fluoride for 30 min. The BCA protein assay kit (Beyotime, Hangzhou, China) was used for total protein quantification. A total of 50 μg of protein was loaded per lane, separated by sodium dodecyl sulfate polyacrylamide gel electrophoresis (SDS-PAGE), and electrophoretically transferred onto polyacrylamide difluoride (PVDF) membranes (Merck Millipore, Bedford, MA, USA). The PVDF membrane was blocked for 1 h with 5% fat-free dry milk in tris-buffered saline with Tween (TBST) and incubated overnight with 1:500 dilution of IL-17A primary antibody (Santa Cruz Biotechnology, CA, USA) and 1:1,000 dilution of glyceraldehyde 3-phosphate dehydrogenase (GAPDH) (Cell Signaling Technology, Danvers, MA, USA). The membrane was then washed 3 times with TBST and incubated for 1 h with a 1:5,000 dilution of the secondary antibody (Beyotime). Finally, immunoreactive proteins were visualized using the ECL reagent (Merck Millipore), and the signals were detected using the ImageQuant Las 4000 mini system (General Electric, Fairfield, CT, USA). The densities of western blotting bands were measured using ImageJ software (National Institutes of Health, Bethesda, MD, USA).

## Statistical analysis

SPSS 21.0 statistical software (IBM, Chicago, IL, USA) and Graphpad prism 6.0 (GraphPad Software, San Diego, CA, USA) were used for statistical analysis. Data are presented as mean ±standard deviation (SD) based on triplicate assays for six independent experiments ($n = 6$). The student's $t$-test was used for comparion between the two groups. Differences between multiple groups were assessed using one-way analysis of variance (ANOVA) method with Tukey's post hoc test. A value of $P < 0.05$ was considered as significant.
## RESULTS

### *P. gingivalis* LPS-treated CD14$^+$ monocytes promote Th17 cell differentiation

To explore the function of CD14$^+$ monocytes on Th17 cell response, CD14$^+$ monocytes treated with 1 μg/mL of *P. gingivalis* LPS were co-cultured with CD4$^+$T cells in the ratios of 1:1, 1:5, and 1:10. The result showed that there was no difference in the mRNA expression of IL-17 and retinoid-related orphan receptor C (RORC) between the *P. gingivalis* LPS-stimulated group and the untreated group in a 1:1 co-culture ratio (Fig. 1A and 1D). However, CD14$^+$ monocytes activated by *P. gingivalis* LPS upregulated the mRNA expression of IL-17 and RORC (Fig. 1B, 1C, 1E and 1F), with the most evident upregulation occurring when cells were cultured in a 1:10 co-culture ratio (Fig. 1C and 1F). Moreover, the frequency of Th17 cells increased in all 1:1, 1:5, and 1:10 cell ratios, especially in the 1:10 (Fig. 1G–1J). Thus, the 1:10 co-culture ratio was used for further analyses to explore the mechanism of in vitro Th17 cell differentiation.

### *P. gingivalis* LPS-stimulated CD14 $^+$ monocytes secrete IL-1 *β*, IL-6, IL-23 and TGF- *β*

To assess the effect of *P. gingivalis* LPS on monocytes, we used 1 μg/mL of *P. gingivalis* LPS to stimulate CD14$^+$ monocytes. There was an upregulation in the mRNA expression of IL-1β, IL-6, TGF-β, and IL-23 (Fig. 2A–2D), and a similar upregulation of the released protein in the supernatants of stimulated CD14$^+$ monocytes was observed relatively to an unstimulated control in the absence of *P. gingivalis* LPS (Fig. 2E–2H).

### Enhancement of Th17 cell differentiation in CD4$^+$T cells by *P. gingivalis* LPS-activated CD14$^+$ monocytes requires cell-cell contact

To confirm the role of soluble cytokines and cell–cell contact in the augmentation of Th17 cell induction by *P. gingivalis* LPS-activated CD14$^+$ monocytes, a transwell assay was used, in which CD14$^+$ monocytes and CD4$^+$T cells were separated by a membrane. The results showed that *P. gingivalis* LPS did not alter the mRNA expression of RORC and IL-17 (Fig. 3A and 3B). Furthermore, there was no difference in the frequency of Th17 cells between the *P. gingivalis* LPS-stimulated group and the untreated group (Fig. 3C and 3D).

### *P. gingivalis* LPS increases the expression of Dll-4 mRNA in CD14$^+$ monocytes

The expressions of Notch ligand Jagged-1 and Dll-4 on APCs have been reported to be associated with Th17 cell differentiation. In the present study, direct cell–cell contact by *P. gingivalis* LPS-activated CD14$^+$ monocytes was required to promote the induction of Th17 cells. Therefore, we explored whether Jagged-1 and Dll-4 were involved in this process. The expressions of Jagged-1 and Dll-4 mRNA were investigated in 1 μg/mL of *P. gingivalis* LPS CD14$^+$ monocytes using qRT-PCR. The results showed that 1 μg/mL of *P. gingivalis* LPS upregulated the expression of Dll-4 mRNA in CD14$^+$ monocytes (Fig. 4A). However, no changes were observed for Jagged-1 mRNA (Fig. 4B).
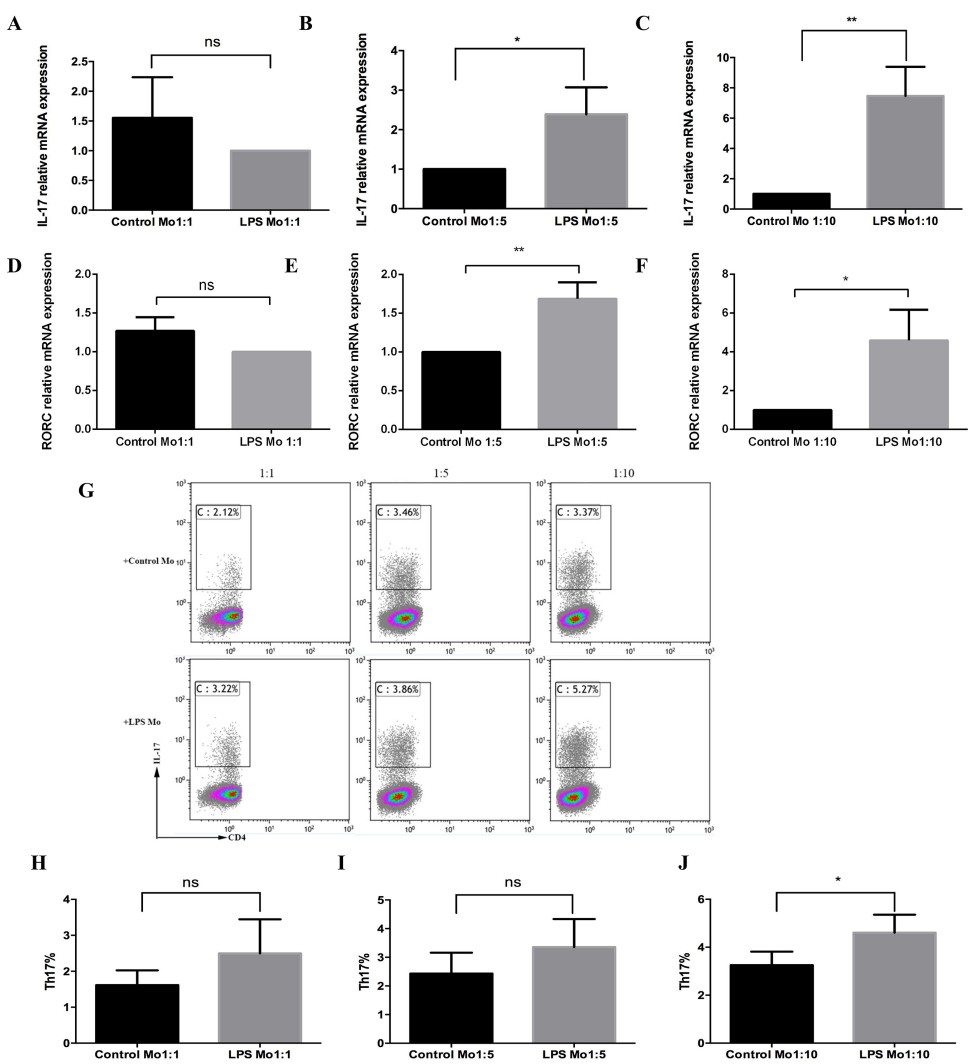

**Figure 1** **Effect of *P. gingivalis* lipopolysaccharide-treated CD14$^+$ monocytes on Th17 cell differentiation.** mRNA expression levels of IL-17 (A-C) and ROR C (D-F) in the presence or absence of *P. gingivalis* lipopolysaccharide were determined using qRT-PCR. (G) Dot plots show the proportion of Th17 cell differentiation determined by using flow cytometry. (H-J) Frequencies of Th17 cells in each of the three co-culture ratios analyzed using flow cytometry. Data are presented as mean ±SD of triplicate assays for six independent experiments ($n = 6$). CD14$^+$ monocytes (Mo), lipopolysaccharide (LPS), retinoid-related orphan receptor C (RORC), no significant difference (ns), * $p < 0.05$, ** $p < 0.01$.

## Th17 cell differentiation induced by *P. gingivalis* LPS decreased in the presence of a blocking Dll-4 antibody

To confirm whether the upregulation of Dll-4 expression on CD14$^+$ monocytes was involved in Th17 cell differentiation, the CD14$^+$ monocytes were pretreated with 5 and 10 µg/mL anti-Dll-4 antibodies for 90 min prior to stimulation with *P. gingivalis* LPS. Flow cytometry demonstrated that the number of differentiated Th17 cells decreased significantly in the 5 and 10 µg/mL Dll-4 blocking groups, compared with the unblocked group (Fig. 5A and 5B). The level of IL-17 protein was determined using western blot, and

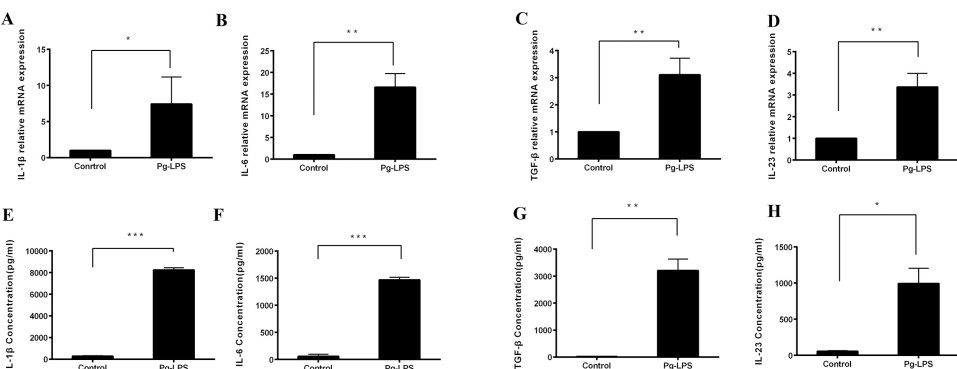

**Figure 2** **Expression of IL-1β, IL-6, TGF-β, and IL-23 in _P. gingivalis_ lipopolysaccharide-treated CD14+ monocytes.** mRNA expression levels of (A) IL-1β (A), IL-6 (B), TGF-β (C), and IL-23 (D) determined using qRT-PCR. Levels of IL-1β (E), IL-6 (F), TGF-β (G), and IL-23 (H) secreted by CD14+ monocytes in the presence or absence of _P. gingivalis_ lipopolysaccharide detected using ELISA. Data are presented as mean ± SD of triplicate assays for six independent experiments ($n = 6$). _P. gingivalis_ lipopolysaccharide (Pg-LPS), * $p < 0.05$, ** $p < 0.01$, *** $p < 0.001$.

it showed a significantly increased level in the _P. gingivalis_ LPS-stimulated group (Fig. 5C and 5D) compared with the _P. gingivalis_ LPS-unstimulated group. Furthermore, compared with the unblocked group, the expression of IL-17 was significantly reduced in the Dll-4 blocking groups, and the decrease was more obvious in the 10 μg/mL Dll-4 blocking group (Fig. 5C and 5D). To analyze the role of anti-Dll-4 in IL-17 secretion, the concentration of secreted IL-17 in the supernatant of the co-cultured cells was determined using ELISA. As expected, the concentration of IL-17 in the 5 and 10 μg/mL Dll-4 blocking groups was significantly downregulated compared with the unblocked group (Fig. 5E). However, no significant difference was found between the 5 and 10 μg/mL Dll-4 blocking groups (Fig. 5E).

## DISCUSSION

Under an inflammatory microenvironment, monocytes can be quickly recruited to the infection sites to participate in an inflammatory process. It has been reported that activated monocytes from rheumatoid arthritis patients specifically induced Th17 cells (_Evans et al., 2009_). A further study showed that optimal Th17 cell induction required the presence of TLR-activated monocytes in vitro (_Evans et al., 2007_). Addition of _P. gingivalis_ to monocyte/CD4+T cell co-cultures promoted a Th17/IL-17 response in a dose- and time-dependent manner (_Cheng et al., 2016_). Nevertheless, the latest study demonstrates that _P. gingivalis_ LPS can directly induce CD4+T cell differentiation to Th17 cells in vitro without APCs (_Zhang et al., 2019_). Therefore, in the present study, to rule out the direct influence of _P. gingivalis_ LPS on CD4+T cells, the CD14+ monocytes activated by _P. gingivalis_ LPS were washed 3 times with PBS to remove excess _P. gingivalis_ LPS before co-cultured with CD4+T cells. The results confirmed that the augmentation of Th17 cells was induced by CD14+ monocytes instead of the residual _P. gingivalis_ LPS contamination. In addition, our results showed that the CD14+ monocytes activated by _P. gingivalis_ LPS increased the frequency

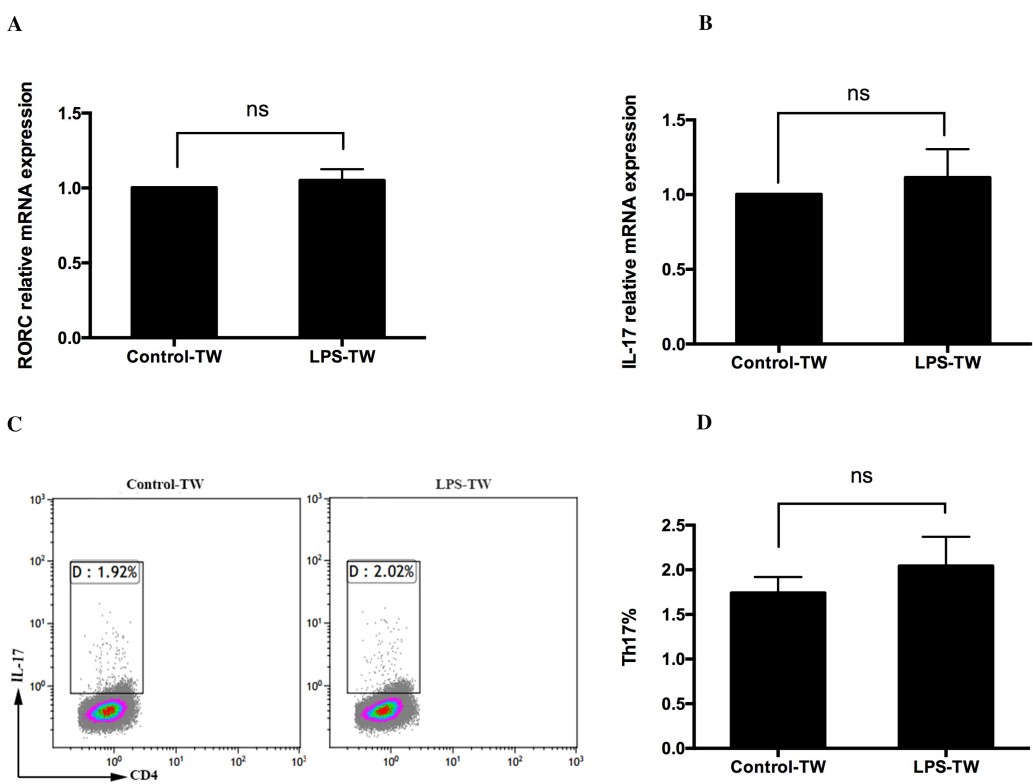

**Figure 3   Role of cell–cell contact in the Th17 cell response induced by *P. gingivalis* lipopolysaccharide-activated CD14$^+$ monocytes.** CD14$^+$ monocytes and CD4$^+$ T cells were cultured in transwells for 3 days, and the expression of RORC (A) and IL-17 (B) mRNA in the presence or absence of *P. gingivalis* lipopolysaccharide was determined. (C) Dot plots show the proportion of Th17 cell differentiation determined using flow cytometry. (D) Frequency of Th17 cells in each group determined by using flow cytometry. Data are presented as mean ± SD of triplicate assays for six independent experiments ($n = 6$). Transwell (TW), lipopolysaccharide (LPS), retinoid-related orphan receptor C (RORC), no significant difference (ns).

of Th17 cells in all 1:1, 1:5, and 1:10 cell ratios, especially in the 1:10, which indicates that the CD14$^+$ monocytes/CD4$^+$T cell ratio also has an impact on the Th17 cell induction. Previous studies have also shown that a monocytes/T cell ratio of 1:1 could effectively induce Th17 cell differentiation (*Cheng et al., 2016*; *Yang et al., 2017*). However, it has been suggested that antigen treated DCs can possess either an immunogenic or tolerogenic function depending on the DC/T cell ratio. High DC/T cell ratios resulted in inhibition of T cell proliferation, while low DCs ratios displayed enhanced T cell-stimulatory properties and supported T cell proliferation (*HöPken et al., 2005*). Therefore, the inhibition of T cell proliferation and immune tolerance induced by high percentage of CD14$^+$ monocytes may be the reason for less frequency of Th17 cells in 1:1and1:5 than 1:10 cell ratios.

To date, IL-1β and IL-6 are prominent cytokine candidates for the polarization of naive Th cells into Th17 cells (*Acosta-Rodriguez et al., 2007*), and TGF-β combined with IL-6 induces RORC to promote Th17 cell differentiation (*Veldhoen et al., 2006*). Although IL-23 is not a necessary factor for Th17 cells, it maintains Th17 cell proliferation (*Zhou*
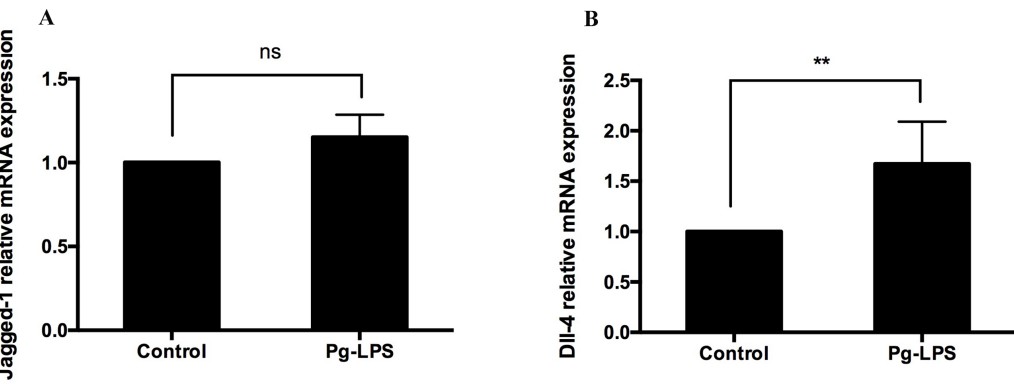

**Figure 4  Expression of Jagged-1 (A) and Dll-4 (B) mRNA in the presence or absence of *P. gingivalis* lipopolysaccharide.** Data are presented as mean ± SD of triplicate assays for six independent experiments ($n = 6$). *P. gingivalis* lipopolysaccharide (Pg-LPS), Delta-like ligand 4 (Dll-4), no significant difference (ns), ** $p < 0.01$.

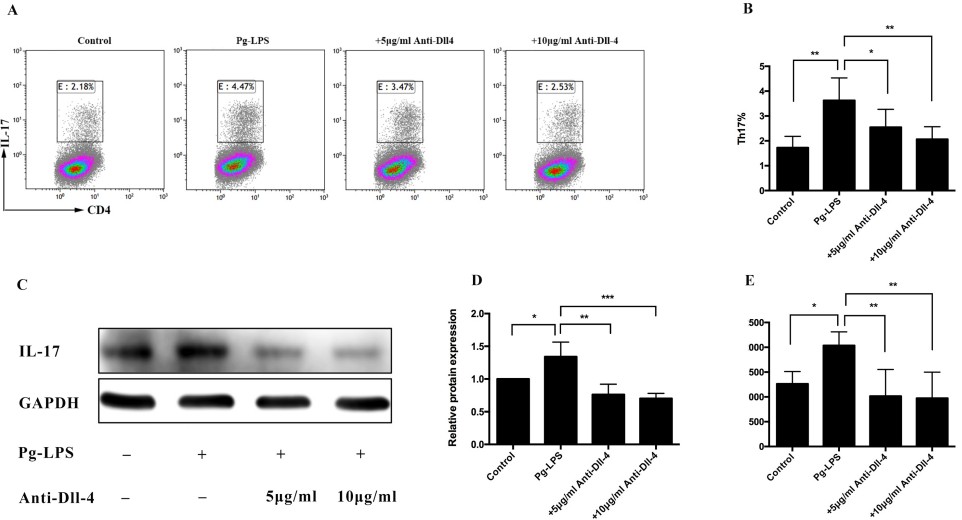

**Figure 5  Role of Dll-4 in the Th17 cell differentiation induced by *P. gingivalis* lipopolysaccharide-activated CD14[+] monocytes.** CD14[+] monocytes were pretreated with 5 and 10 μg/mL anti-Dll-4 antibody, for 90 min prior to addition of the *P. gingivalis* lipopolysaccharide. The cells were then cocultured with the activated CD4[+] T cells for 5 days. (A) Dot plots show the proportion of Th17 cell differentiation determined using flow cytometry. (B) Frequency of Th17 cells in each group determined using flow cytometry. (C) Protein level of IL-17 detected using western blotting. GAPDH was used as the protein loading control. (D) The intensified bands of IL-17 were measured using ImageJ software and were normalized to GADPH. (E) IL-17 concentration in the supernatant of the co-cultured cells measured after 5 days using ELISA. Data are presented as mean ± SD of triplicate assays for six independent experiments ($n = 6$). *P. gingivalis* lipopolysaccharide (Pg-LPS), Delta-like ligand 4 (Dll-4), * $p < 0.05$, ** $p < 0.01$, *** $p < 0.001$.

*et al., 2007*). *Segura et al. (2013)* blocked IL-1β, IL-6, IL-23, and TGF-β and observed a decrease in the differentiation ratio of Th17 cells. *Moutsopoulos et al. (2012)* reported an upregulation of IL-1β and IL-6 associated with *P. gingivalis* stimulated-myeloid APCs.

In the present study, *P. gingivalis* LPS-stimulated CD14$^+$ monocytes secreted IL-6, IL-1 $\beta$, TGF-$\beta$, and IL-23 proteins. Moreover, the mRNA expression of these cytokines was obviously upregulated, thus indicating that the CD14$^+$ monocytes activated by *P. gingivalis* LPS might be related to Th17 cell response. To explore the function of cytokines induced by the *P. gingivalis* LPS in Th17 cell response, the transwell assay was used. However, no difference was observed in Th17 cell differentiation in the presence or absence of *P. gingivalis* LPS when cells were cultured separately. Moreover, the ratio of Th17 cells was significantly lower than in direct co-culture, thus indicating that the CD14$^+$ monocytes activated by *P. gingivalis* LPS failed to induce Th17 cell response in the absence of direct cell–cell contact. It can be concluded that cell–cell contact may play pivotal roles in Th17 cell differentiation. Besides, we also simultaneously observed that the cytokine concentrations stimulated by 1 $\mu$g/mL of *P. gingivalis* LPS-activated CD14$^+$ monocytes were considerably lower than Th17 skewing conditions in vitro. This might be another reason why *P. gingivalis* LPS could not effectively induce Th17 cell differentiation in the absence of direct cell–cell contact. In summary, these results indicate that the induction of Th17 cell differentiation is not solely cytokine-dependent but rather requires cell–cell contact with *P. gingivalis* LPS activated monocytes.

It has been reported that cell–cell contact signal, such as adhesion, co-stimulation factors expressed on CD14$^+$ monocytes play essential roles in Th17 cell differentiation (*Yang et al., 2017*; *Roberts, Dickinson & Taams, 2015*). Recent research has shown that the Notch signaling pathway plays a pivotal role in T cell lineage commitment and is involved in the regulation of Th17 cell response (*Amsen, Helbig & Backer, 2015*; *Jiao et al., 2014*; *Keerthivasan et al., 2011*). It is well known that the interaction between the Notch receptors and ligands is the starting point for the Notch signaling cascade. Notch ligands expressed on the surface of APCs are associated with Th17 responses, and Jagged-1 and Dll-4 are the most widely studied ligand (*Higashi et al., 2010*; *Weng et al., 2017*; *Wang et al., 2015*). In the present study, CD14$^+$ monocytes were activated by *P. gingivalis* LPS, and the expressions of Jagged-1 and Dll-4 were determined. The results showed that 1 $\mu$g/mL *P. gingivalis* LPS upregulated the expression of Dll-4 mRNA but did not affect Jagged-1 mRNA. Higashi's study showed that curdlan stimulation induces an increased level of Jagged-1 mRNA but not Dll-4 in Mo-DCs (*Higashi et al., 2010*). *Tsao et al. (2011)* reported increased Jagged-1 on the surface of the macrophages exposed to LPS. In this study, *P. gingivalis* LPS increased the expression of Dll-4 but not Jagged-1. The discrepancy between different studies may be due to the use of different stimuli and may be due to cell specificity.

Dll-4-mediated Notch signaling is thought to play a pivotal role in the regulation of Th17 cell response (*Jiang et al., 2015*). The present study showed *P. gingivalis* LPS upregulated Dll-4 expression on CD14$^+$ monocytes. Moreover, *P. gingivalis* LPS enhanced Th17 cell differentiation by cell–cell contact. On the basis of these results, we speculate that the Dll-4 expressed on CD14$^+$ monocytes correlates with Th17 cell response in the periodontal inflammatory microenvironment. Thus, the mechanism of Th17 cell differentiation was analyzed using a blocking Dll-4 antibody. The results showed that Dll-4 blockade reduced the frequency of Th17 cells, and the IL-17 protein level in the supernatant. These results are consistent with the findings of Mukherjee, who reported

that the TLR-mediated stimulation of IL-17 responses depends on the stimulus and that Dll-4 protein enhances T cell skewing toward IL-17 production (*Mukherjee et al., 2009*). Previous studies have reported that blocking Dll-4-Notch signaling in animal models of experimental autoimmune encephalomyelitis and asthma decreased both Th17 response and clinical symptoms severity (*Weng et al., 2017*; *Bassil et al., 2011*). Therefore, Dll-4 may present a promising therapeutic target for immunomodulation in the development of periodontitis and deserves further exploration in vivo.

## CONCLUSIONS

In summary, we demonstrated that *P. gingivalis* LPS activated CD14$^+$ monocytes promote Th17 cell response in a way that requires cell–cell contact. This upregulation is partially reversed in the presence of anti-Dll-4, thus indicating that the Notch ligand Dll-4 expressed on the CD14$^+$ monocytes is involved in Th17 cell response.

### Funding
This study was supported by the National Nature Science Foundation of China (grant numbers 81870770 and 81300887). The funders had no role in study design, data collection and analysis, decision to publish, or preparation of the manuscript.

### Grant Disclosures
The following grant information was disclosed by the authors:
The National Nature Science Foundation of China: 81870770, 81300887.

### Competing Interests
The authors declare there are no competing interests.

### Author Contributions
- Chi Zhang performed the experiments, prepared figures and/or tables, and approved the final draft.
- Chenrong Xu analyzed the data, prepared figures and/or tables, and approved the final draft.
- Li Gao, Xiting Li and Chuanjiang Zhao conceived and designed the experiments, authored or reviewed drafts of the paper, and approved the final draft.

### Human Ethics
The following information was supplied relating to ethical approvals (i.e., approving body and any reference numbers):
The Ethics Committee of the Hospital of Stomatology of Sun Yat-Sen University approved this research (Ethical Application Ref:ERC-2014-08).

### Data Availability
Raw measurements are available in the Supplemental Files.

## Supplemental Information

Supplemental information for this article can be found online at http://dx.doi.org/10.7717/peerj.11094#supplemental-information.

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
