# Peer review of "Porphyromonas gingivalis lipopolysaccharide promotes T-hel per17 cell differentiation by upregulating Delta-like ligand 4 expression on CD14+ monocytes"

_PeerJ, doi:10.7717/peerj.11094_

## Round 0.1 · original submission · Major Revisions

Dear Dr. Zhang and colleagues:

Thanks for submitting your manuscript to PeerJ. I have now received two independent reviews of your work, and as you will see, the reviewers raised some substantial concerns about the research. Despite this, these reviewers are optimistic about your work and the potential impact it will have on research studying immune responses to Porphyromonas gingivalis lipopolysaccharide. Thus, I encourage you to revise your manuscript, accordingly, taking into account all of the concerns raised by both reviewers.

Please also not that reviewer 2 has included a marked-up version of your manuscript.

In your revision, please provide a rationale for why you are not directly measuring cytokine expression. Also, please address the statistical shortcomings identified by reviewer 2.

I look forward to seeing your revision, and thanks again for submitting your work to PeerJ.

Good luck with your revision,

-joe

Reviewer 1 ·

Basic reporting

This article meets the standard.

Experimental design

This study need to determine many cytokines but, the method which was used ie, PCR or western blot were not adequate to prove the cytokine activity.

I suggest to change the cytokine determination method to be ELISA.

Validity of the findings

According to the cytokine determination methods were not suitable so, the results in this part still not adequate to conclude.

Additional comments

In general, this article is very interesting in the field of inflammation and oral disease. If the author can improve the point as my suggestion, it will increase validity of this study.

Reviewer 2 ·

Basic reporting

1. The manuscript should be sent to proofed reader who is a native speaker because I found some grammatically errors, but they are not a big issue. Such as line 224 (relatively instead of relative), line 270 (Nevertheless, the latest instead of Nevertheless, latest)

2. Please rearrange the methods in the abstract. The authors mentioned mRNA expression lastly in the abstract (line 48), but it is not the last method that you used in your experiment and also in your manuscript.

3. Please explain more about the roles of Th17 in the progression or severity of periodontal diseases to highlight the importances of your finding.

4. Please add the full name of RORC when the authors first mentioned it (line 215). They first mentioned the full name of RORC at line 513 which is in the figure legends.

5. Please add the full name of NIH (National Institutes of Health) (line 200).

6. The authors should recheck the reference format such as line 378 and 420.

Experimental design

1. The authors should divide the co-culture from Transwell assays (non-contact co-culture). When they included both experiments in one topic, it's difficult to understand that those are separate experiments. And also, it will be highlight the impact of contact co-culture to the result.

2. Please explain more detail why did the authors add PMA, ionomycin and brefeldin A for the last 5 h of contact co-culture between monocyte and T cells (line 146).

3. For line 182 and 183, please mention the cytokines in numbering order (IL-1B, IL-6, IL-17A, IL-23 and TGF-B) as mentioned at line 220 and 291.

4. Please add the state name of Cell singling Technology company (line 195).

5. The authors should also highlight that the CD14+ monocyte cells were washed 3 times with PBS to remove excess LPS before co-culture with CD4+ T cells in material and method section, not only in the discussion part (line 273-274) .

Validity of the findings

The most important thing that the authors should be concerned is statistical analysis that they have used. For example,

1. When the author has mentioned that they set the significant level at p<0.05. That's enough and there is no need to show the significant level at p<0.01 or p<0.001.

2. The authors should mention that they used T-test when they compared the mRNA expression between unstimulated (control) and LPS stimulated group (Figure 1) only between the same proportion. Because when they demonstrate many proportions (6 bars) in one graph, the audience will be confused about how to interpret the data.

3. The authors should also mention that ANOVA is used to demonstrate the significance when there are more than 3 groups in the same experiment. When the authors found that ANOVA showed any significant, they should further run the post-hoc comparison, to compare the differences between groups or the difference between any test group to control (figure 5).

4. Please add more detail about is there any study that demonstrated the same result as your study (increased in expression only for Dll-4, but not Jagged-1, line 324).

Additional comments

-

Annotated reviews are not available for download in order to protect the identity of reviewers who chose to remain anonymous.

---

## Round 0.2 · accepted · Accept

Dear Dr. Zhang and colleagues:

Thanks for revising your manuscript based on the concerns raised by the reviewers. I now believe that your manuscript is suitable for publication. Congratulations! I look forward to seeing this work in print, and I anticipate it being an important resource for groups studying immune responses to Porphyromonas gingivalis lipopolysaccharide. Thanks again for choosing PeerJ to publish such important work.

Best,

-joe

Reviewer 1 ·

Basic reporting

This manuscript meet the standard of the journal in general.

Experimental design

The experimental design is able to address the research question.

Validity of the findings

This study provides a novel findings with standard validity of results and conclusion.

Additional comments

This manuscript has been improved in all issues that suggested by the reviewers.

Reviewer 2 ·

Basic reporting

After revision, this article meets the standards.

Noted: please correct the spelling of the word "background" in the abstract.

Experimental design

After revision, the design and methods are much more clearer than the first version. I really appreciate for the authors contribution.

Validity of the findings

After revision, this article meets the standards.

Additional comments

Overall, I really appreciate for your effort to improve the quality of the manuscript. The revised version is really impressive. Thank you very much for your time and contribution.